

# Competitive interactions between corals and turf algae depend on coral colony form

Thomas Swierts[1,2] and Mark JA Vermeij[2,3]

[1] Marine Biodiversity, Naturalis Biodiversity Center, Leiden, the Netherlands
[2] Aquatic Microbiology, Institute for Biodiversity and Ecosystem Dynamics, University of Amsterdam, Amsterdam, the Netherlands
[3] Carmabi Foundation, Willemstad, Curaçao

## ABSTRACT

Turf algae are becoming more abundant on coral reefs worldwide, but their effects on other benthic organisms remain poorly described. To describe the general characteristics of competitive interactions between corals and turf algae, we determined the occurrence and outcomes of coral–turf algal interactions among different coral growth forms (branching, upright, massive, encrusting, plating, and solitary) on a shallow reef in Vietnam. In total, the amount of turf algal interaction, i.e., the proportion of the coral boundary directly bordering turf algae, was quantified for 1,276 coral colonies belonging to 27 genera and the putative outcome of each interaction was noted. The amount of turf algal interaction and the outcome of these interactions differed predictably among the six growth forms. Encrusting corals interacted most often with turf algae, but also competed most successfully against turf algae. The opposite was observed for branching corals, which rarely interacted with turf algae and rarely won these competitive interactions. Including all other growth forms, a positive relationship was found between the amount of competitive interactions with neighboring turf algae and the percentage of such interaction won by the coral. This growth form dependent ability to outcompete turf algae was not only observed among coral species, but also among different growth forms in morphologically plastic coral genera (*Acropora, Favia, Favites, Montastrea, Montipora, Porites*) illustrating the general nature of this relationship.

## INTRODUCTION

Benthic algae and corals are among the main groups competing for space on coral reefs (*Lang & Chornesky, 1990*; *Karlson, 2002*; *Fong & Paul, 2010*) and anthropogenic stressors have led to an increase of the former at the cost of the latter (*Hughes, 1994*; *Bellwood et al., 2004*; *Hoegh-Guldberg et al., 2007*). Favorable conditions for algal growth are created by the reduced abundance of herbivorous fish due to overfishing and eutrophication resulting from the unsustainable use of coastal areas (e.g., *Hughes, 1994*; *Pandolfi et al., 2003*; *Pandolfi et al., 2005*). As algae increase in abundance, they can actively overgrow live corals or passively take over space after corals have died. Feedback processes exacerbate the decline of coral populations as algae provide refuges for coral pathogens and algal exudates fuel

Corresponding author
Thomas Swierts, tswierts@gmail.com, thomas.swierts@naturalis.nl

bacterial sources of coral mortality (e.g., *Kline et al., 2006*; *Smith et al., 2006*; *Rohwer & Youle, 2010*).

Many coral reefs have seen large increases in the benthic cover of turf algae, a less noticeable and more complex functional group than the more often studied macroalgae. Turf algae (or "algal turfs") are dense, multi-species assemblages of filamentous benthic algae, including small individuals of macroalgae and cyanobacteria, that are typically less than 1 cm in height (*Connell, Foster & Airoldi, 2014*). The general absence of turf algae in studies of coral reef ecology and conservation is paradoxical because algal turfs are or are becoming one of the most abundant benthic groups typical of degrading reef communities (*Littler, Littler & Brooks, 2006*; *Sandin et al., 2008*). Compared to other algal groups such as macroalgae and crustose coralline algae (CCA), turf algae occupy available space quicker (*Diaz-Pulido & McCook, 2002*), grow faster (*Littler, Littler & Brooks, 2006*) and are less vulnerable to grazing and water turbulence (*Hay, 1981*; *Cheroske, Williams & Carpenter, 2000*). Turf algae can weaken or overgrow and kill neighboring corals, though the particular outcome of a competitive interaction depends on the species involved (*Jompa & McCook, 2003*) and the environmental setting in which the interaction takes place (*Vermeij et al., 2010*; *Barott et al., 2012*).

Sessile organisms, like corals, have developed an array of physical and chemical defensive mechanisms against pathogens and predators. These defenses come at a cost since the allocation of resources towards protection reduces those available for growth and reproduction (*Herms & Mattson, 1992*; *Endara & Coley, 2011*; *Züst et al., 2011*). Reduced growth in response to competitive interaction has been demonstrated in terrestrial plants (*Züst et al., 2011*) as well as marine sponges (*Leong & Pawlik, 2010*) and the trade-off between fast growth and defense has been a topic of interest in the biology of sessile organisms for decades (*Coley, Bryant & Chapin, 1985*; *Herms & Mattson, 1992*; *Endara & Coley, 2011*). The resource availability hypothesis (RAH), originally proposed for terrestrial plants, can potentially be used for benthic phototrophs and states that the costs of allocating resources away from growth to defenses are relatively higher for fast growers than for slow growers (*Endara & Coley, 2011*). For slow growing corals, this implies that tissue loss due to predation or competition is more difficult to compensate by regenerative or faster growth, making investing energy in defenses worthwhile. For fast growing corals, the investment in defenses would have a negative effect on growth required to escape competition and therefore these corals may be less inclined to invest energy in defenses.

Corals are known for their morphological plasticity and wide variety of growth forms, from encrusting to heavily branched. Branching growth forms are typically fast growing species (e.g., *Yap, Alino & Gomez, 1992*) that extend above the benthos allowing them to avoid interactions with neighboring organisms, including turf algae. In contrast, slow growing and non-erect growth forms (e.g., massive- and encrusting growth forms) are less likely to escape such interactions with neighboring algae. Rather than avoiding competitive interaction through upward growth, such species are expected to actively fight off their opponents (e.g., through the production of secondary metabolites) to survive (*Lang & Chornesky, 1990*; *Karlson, 2002*).

With the increasing abundance of algal turfs on coral reefs (e.g., *Littler, Littler & Brooks, 2006*; *Sandin et al., 2008*), it is important to study their interactions with corals and look for general patterns that allow predictions on how coral communities might change in the future. The use of morphological variability in corals as a predictive factor determining the outcome of competitive interactions, i.e., whether a coral wins or loses the interaction, with turf algae could be considered in this context and the usefulness of such approach has already been proven in marine sponges (*Bell & Barnes, 2003*). Following the expectations of the RAH, corals with morphologies associated with fast growth (e.g., branching corals) are less likely to win competitive interactions with turf algae compared to slower growing species that, according to the RAH, would have more resources and/ or mechanisms available to successfully compete with neighboring turf algae.

In this study we tested for differences among six common coral growth forms (i.e., branching, encrusting, massive, plating, solitary and upright) in terms of the occurrence of turf algal interaction along their edges and their success in ''winning'' these interactions. We hypothesized that faster growing species characterized by erect growth forms (i.e., branching, upright) interacted with turf algae along a smaller part of their perimeter compared to slower growing growth forms (i.e., massive, encrusting). Secondly, we hypothesized that growth forms associated with slow growth (i.e., encrusting, massive) would win aforementioned interactions more often than fast growers following the predictions of the RAH.

## METHODS

### Site description

This research was carried out in Ninh Van Bay (12.356°N; 109.277°E), part of the South Chinese Sea (Fig. 1A) and located nine kilometers northeast of Nha Trang, the seventh largest city of Vietnam. All surveys were conducted between March and April 2013. The reef at our study location extended over approximately 500 meter parallel to the wave-sheltered eastern side of Ninh Van Bay between depths of zero and five meters (Fig. 1B). Fishing is prohibited at this site, but occurs in other sections of Ninh Van Bay and adjacent waters (*Ngoc, Flaaten & Anh, 2009*).

### Benthic cover

Photoquadrats were used to quantify the composition of the benthic community (*Preskitt, Vroom & Smith, 2004*). Four transects of 50 m were deployed with at least 50 m in between at a depth between two and five meters. Along each transect, 25 quadrats (0.9 × 0.6 m) were laid down at 2 m intervals and subsequently photographed using a digital underwater camera (Nikon AW100 Coolpix). Benthic cover and composition of all major functional groups were analyzed underneath 100 randomly placed points overlain on each picture using Coral Point Count with Excel extensions (*Kohler & Gill, 2006*). Corals and macroalgae were identified to genus-level whereas CCA and turf algae were classified as individual functional groups. Other benthic organisms (e.g., sponges, soft corals) were rare at our study site and not detected in our surveys. Non-biological substrates (e.g., sand, rubble, dead coral) were specified as such.
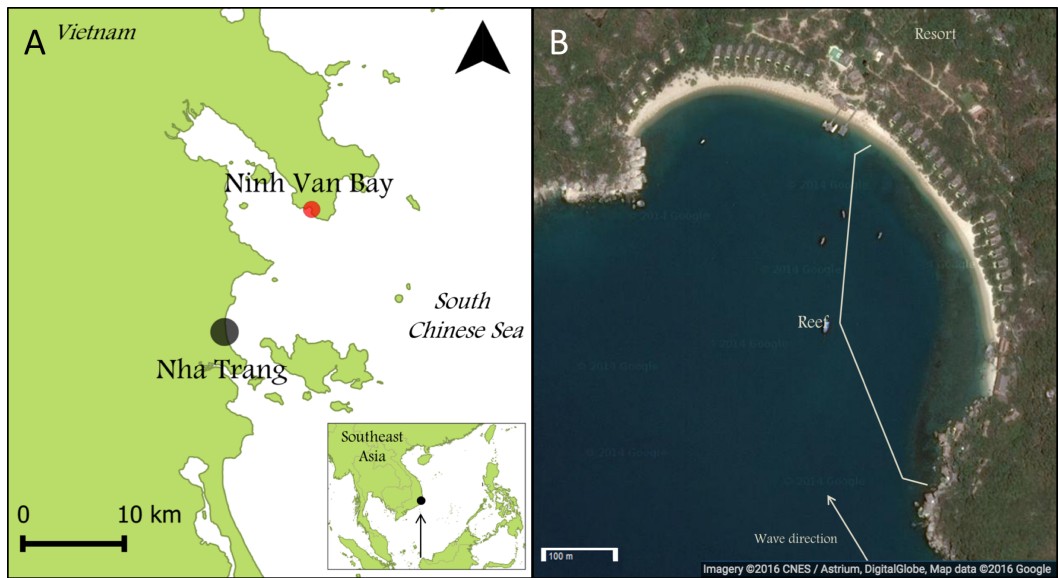

**Figure 1** **Maps of Nha Trang, Ninh Van Bay and study site.** (A) The study site is indicated by the red marker (12.356°N; 109.277°E). (B) All surveys were conducted on the wave-sheltered southeastern side of the reef marked by the white lines. (Map credit: Google, DigitalGlobe).

## Surveys of coral algae interactions

To study coral–algal interactions, we used a line intercept approach described by *Barott et al. (2009)* and *Barott et al. (2012)*. Along a depth range between two and five meter forty-two transects (25 m) were haphazardly positioned across the reef in various directions so that individual transects never bisected others. Each coral colony on each transect was photographed against a 30 cm ruler for scale. A top view photo was taken along with photos from various directions and distances to capture the entire coral–algal boundary. The proportion of coral border involved in each type of coral–algal interaction was later measured in ImageJ 1.48 (*Abramoff, Magalhaes & Ram, 2004*) using the top view photo, whereby side photos and close-ups were used to confirm the initial assessment if necessary. The putative outcome of each interaction was estimated by eye from the same pictures (see below). Algae were classified to genus for macro algae and to a single functional group for turf algae and CCA. The only exception to this method was made for branching *Acropora* corals since colony bases were generally hidden under an entanglement of branches. These *Acropora* colonies could exist of more than 100 individual branches. For colonies with more than 40 branches we photographed 40 individual primary branches and used the percentage of branches with algal growth as a proxy for the percentage of the coral border involved in competitive interactions. For smaller colonies, we analyzed all individual primary branches for algal growth. Branches that were completely overgrown from the primary branch upwards were considered dead and not included in the calculations.

For all colonies interacting with neighboring algae, the putative outcome of each competitive interaction was noted following the classifications of *Barott et al. (2012)*. In short, competitive outcomes were classified as: (I) coral outcompeted neighboring algae, (II) algae outcompeted neighboring coral, or (III) if there was no obvious ''winner'' the

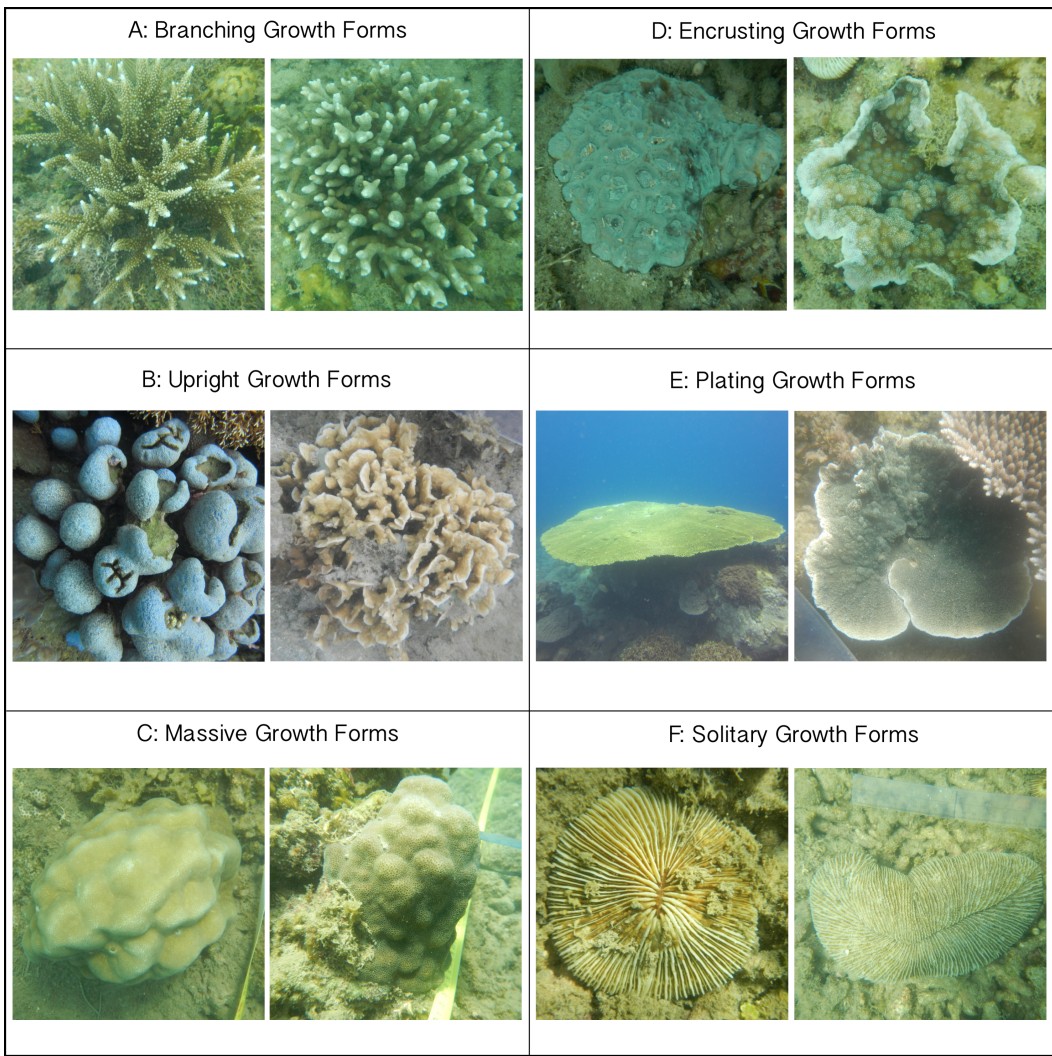

**Figure 2** **Images of classified growth forms.** (A) Branching Growth Forms (most dominant genera: *Acropora n* = 234, *Porites n* = 76, Pocillopora *n* = 66); (B) Upright Growth Forms (*Porites n* = 39, *Pavona n* = 36); (C) Massive Growth Forms (*Porites n* = 98, *Favia n* = 54, *Favites n* = 31); (D) Encrusting Growth Forms (*Galaxea n* = 57, *Porites n* = 53, *Montipora n* = 48); (E) Plating Growth Forms (*Acropora n* = 70, *Montastrea n* = 6); (F) Solitary Growth Forms (*Fungia n* = 69, *Ctenactis n* = 11). All photos are taken by Thomas Swierts.

interaction was classified as neutral. When healthy coral tissue was growing over the algal turfs, corals were assumed to outcompete neighboring algae (see also Fig. 2 in *Barott et al. (2012)*). Algae were regarded as winning the competitive interaction when the coral colony's edge suffered from bleaching, discoloration, tissue necrosis or when algal turfs overgrew the coral surface. The first two categories are considered '*directional competitive interactions*,' whereby one organism overgrew or killed its neighbor and eventually would take its place. Note that a single coral colony could be involved in multiple competitive interactions with multiple algal genera or functional groups and that each interaction could have multiple competitive outcomes. Each interaction and their outcomes were included proportionally in our analyses (Supplemental Information). We stress that we focused on

describing the short-term dynamics of interaction zones and not on the long-term fate of individual colonies. Lastly we measured the maximum coral diameter to investigate a potential relationship between coral colony size and competitive outcome.

## Definitions of colony growth forms and size classes

Every coral colony was classified into one of six growth forms; branching (B), encrusting (E), massive (M), plating (P), solitary (S) and upright (U) (Fig. 2). These classifications are based on commonly used typologies of coral growth forms (e.g., *McCook, Jompa & Diaz-Pulido, 2001*; *Muko et al., 2013*), that were binned to represent the morphological variation at our study site. Corals showing clear digitate-, corymbose- or branched patterns were grouped as *branching* (Fig. 2A), whereas all vertical orientated corals lacking these patterns (e.g., columnar- and foliating corals) were classified as *upright* (Fig. 2B). *Massive* corals were hemispherically shaped (Fig. 2C). Both *plating* and *encrusting* corals grew horizontally, whereby the former were elevated above the benthos and the latter grew over the benthos (Figs. 2D and 2E). Lastly, *solitary* corals were non-attached, free-living coral colonies moving over the top of the substratum (Fig. 2F) and in this study only included the genera *Fungia* and *Ctenactis*. All colonies were categorized into one of six size classes ('0–5 cm,' '5–10 cm,' '10–20 cm,' '20–40 cm,' '40–80 cm,' '80+ cm') following *Barott et al. (2012)*. For branching *Acropora* colonies it was not possible to identify individual colonies in dense *Acropora* thickets so that this group was excluded from the size class comparisons.

## Statistical analyses

We compared the general abundance of algal groups to their relative abundance in coral–algal interactions using a two-tailed binomial test to test whether certain algal groups were over- or underrepresented in coral–algal interactions. Non-parametric, multiple comparisons Kruskal–Wallis tests with Bonferroni corrections (to compensate for multiple comparisons) were used to test for differences in the average proportion of coral edge interacting with turf algae for each of the six growth forms and the six size classes. Non-parametric multiple comparisons Kruskal–Wallis tests were also used to test which different growth forms and different size-classes were more successful in winning competitive interactions with turf algae. For these last tests we used the fraction of corals winning the interaction divided by the total amount of directional competitive interactions (average % of corals winning / (average % of corals winning + average % of algae winning)). Aforementioned tests were not only performed on growth morphology whereby different taxa were binned into one morphological category, but also within morphologically plastic coral genera that harbored multiple growth forms. We only compared growth forms within a single genus, if the growth forms were represented by at least 20 individuals each. Based on this comparison we could detect variation in the performance amongst growth forms without the possibility of falsely accrediting the differences to genus-specific reactions to turf algae. All statistical analyses were conducted in R (*R Development Core Team 2010*).

**Table 1  Benthic cover and composition of the coral colony border.** Proportion of the total benthos and coral colony edge covered by or competing with different algal groups.

| | Coral border interacting (cm) of 50,538 cm total coral border | Coral border interacting (%) | Average interaction along coral edge per colony (%) | Coverage of reef benthos (%) | p-value |
|---|---|---|---|---|---|
| Brown algae | 1,699 | 3.4 | 3.0 | 5.0 | *** |
| Green algae | 147 | 0.3 | 0.3 | 6.5 | *** |
| Red algae | 1,136 | 2.2 | 2.6 | 0.4 | *** |
| Turf algae | 21,066 | 41.7 | 42.8 | 34.9 | *** |
| Total with algae | 24,049 | 47.6 | 48.7 | 46.8 | |

Notes.

p-values indicate significance levels of the disproportionality between the quantity of the functional group along the coral colony border compared to its coverage of the reef benthos based on a two-tailed binomial test: $* \leq 0.05$; $** \leq 0.01$; $*** \leq 0.001$.

## RESULTS

### Benthic cover and coral–algal interactions

The reef community of Ninh Van Bay was comprised of macro- and turf algae (46.8%), stony corals (37.5%) and the remaining 15.7% consisted of non-biological substrates. The algal community was dominated by turf algae (74.6% of the total algal cover) accounting for 34.9% of the total reef community. After turf algae, green- (6.5%; *Dictyosphaeria* spp.; *Halimeda* spp. *Valonia* spp.), brown-(5.0%; *Dictyota* spp., *Padina* spp., *Sargassum* spp., *Turbinaria* spp.) and red algae (0.4%; *Amphiroa* sp., crustose coralline algae) were the most dominant algal taxa respectively (Table 1).

The circumferences of the 1,046 measurable coral colonies (i.e., all coral colonies except *Acropora* with a branching growth form) added up to 50,538 cm, of which 24,049 cm (47.6%) was in direct contact with algae. Again, turf algae were the dominant algal group, accounting for 87.6% of the total algal community along these coral borders. In the 230 branching *Acropora* coral colonies an estimated 47.0% of all branches were interacting with algae, of which 91.2% were algal turfs.

### Coral growth form and coral–algal interactions

Of the six growth forms, encrusting corals were most commonly engaged in competitive interactions with turf algae (along 79.3% of their edges; Fig. 3A). In terms of the occurrence of competitive interactions, encrusting corals were followed by massive-(59.1%), upright-(49.7%) and branching corals (27.3%). Plating-(12.0%) and solitary corals (12.4%) experienced the least amount of turf algal interaction of the six growth forms (Fig. 3). Encrusting corals always had more turf algae growing along their borders than all other growth forms, i.e., from 1.34 times more than massive corals up to 6.61 times more than plating corals. Massive and upright coral colonies have more competitive interactions with turf algae than branching, plating and solitary growth forms (Table 2). Only in a few occasions did different morphologies experience a similar amount of coral–algal competitive interaction, i.e., branching and solitary, plating and solitary and upright and massive colonies (Table 2).

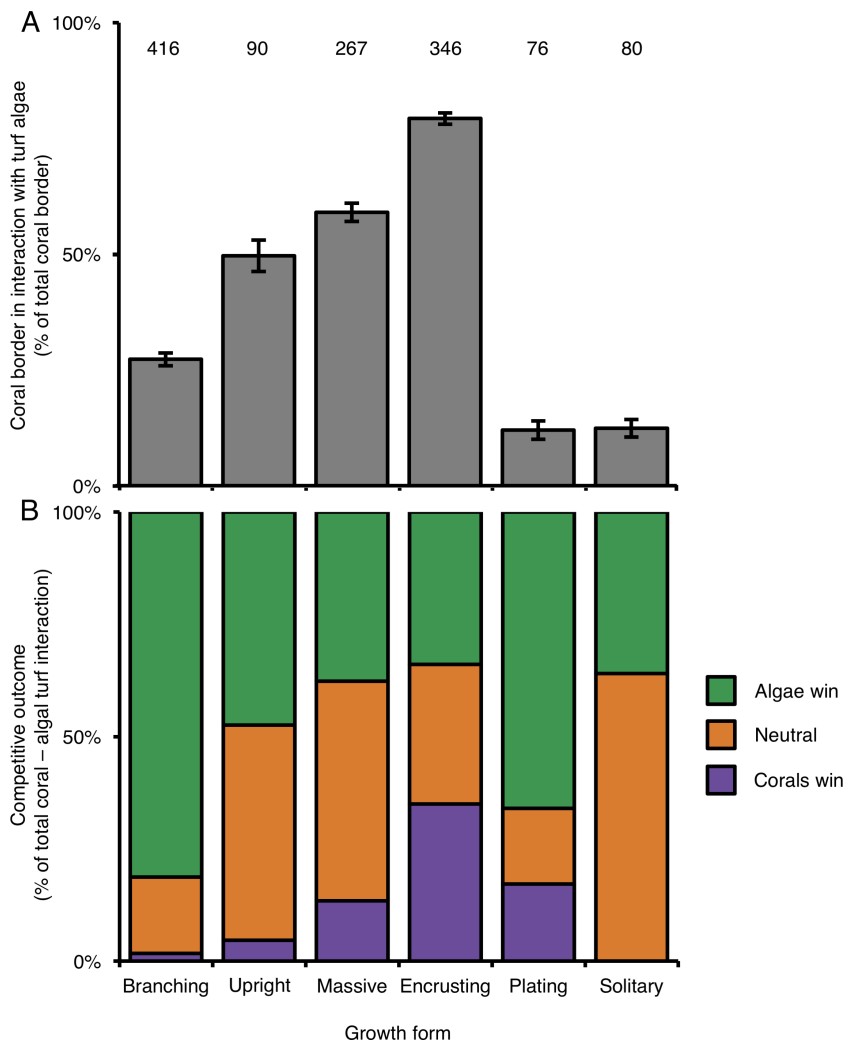

**Figure 3  Competitive interactions between corals and algae per coral colony growth form.** (A) Amount (%) of coral colony border interacting with turf algae per coral colony growth form. Error bars indicate standard error; numbers indicate the amount of samples. (B) Competitive outcomes per coral colony growth form. Purple indicates the proportion of corals winning, orange indicates the proportion of neutral interactions and green indicates the proportion of algae winning.

## The effect of coral growth form on competitive outcomes

Similar to above, a colony's success in outcompeting neighboring turf algae depended on its growth form. Encrusting corals were the most successful competitors against turf algae and won 35% of the interactions (Fig. 3B), followed by plating-(17%), massive-(13%), upright-(5%) and branching corals (2%). Solitary corals were only observed in neutral interactions or being outcompeted by turf algae (Fig. 3B). Turf algae were least successful in winning competitive interactions with encrusting corals (35%) and were increasingly better at outcompeting neighboring corals for solitary-(36%), massive-(38%), upright-(47%), plating-(66%), and branching growth forms (82%) (Fig. 3B).

Overall, turf algae won competitive interactions with corals more often than vice versa (Fig. 4), but growth forms that experienced more competitive interactions with turf algae

**Table 2  Pairwise comparisons between different coral colony growth forms with the Kruskal–Wallis test.** Levels of significance of pairwise comparisons between coral colony growth forms for (I) different quantities of coral-algal competition along coral borders (in row under 'Border') and (II) different competitive outcomes (in row under 'Competitive outcome').

| Growth forms | Border | Competitive outcome |
| --- | --- | --- |
| Branching–Upright | *** | n.s. |
| Branching–Massive | *** | *** |
| Branching–Encrusting | *** | *** |
| Branching–Plating | * | n.s. |
| Branching–Solitary | n.s. | n.s. |
| Upright–Massive | n.s. | n.s. |
| Upright–Encrusting | *** | *** |
| Upright–Plating | *** | n.s. |
| Upright–Solitary | *** | n.s. |
| Massive–Encrusting | *** | *** |
| Massive–Plating | *** | n.s. |
| Massive–Solitary | *** | ** |
| Encrusting–Plating | *** | *** |
| Encrusting–Solitary | *** | *** |
| Plating–Solitary | n.s. | n.s. |

**Notes.**
The number of asterisks indicate the $p$-values after Bonferroni correction: * $\leq 0.05$; ** $\leq 0.01$; *** $\leq 0.001$, n.s. = not significant.

were also more likely to be successful during such interactions. To illustrate, encrusting corals experienced the highest average amount of turf algal interaction along their perimeter, but they were the most successful growth form competing against turf algae, winning 49% of all competitive interactions (Fig. 4). The opposite pattern was observed for branching corals that had a relatively low number of their branches partly covered by turf algae, but could only be qualified as winning in 3% of all competitive interactions.

## Taxon or growth form as the main driver of competitive success?

For six coral genera (*Acropora*, *Favia*, *Favites*, *Montastrea*, *Montipora* and *Porites*) at least twenty individuals of more than one growth form could be found. Comparing coral–turf algal interactions among growth forms within individual genera again showed differences in competitive outcomes among growth forms within the same genus (Fig. 5). *Acropora* was the most abundant coral genus on the reef (79% of total coral cover) and branching corals in this genus experienced three times more competitive interactions with turf algae along there edges than plating *Acropora*'s (Fig. 5A). Corals of the genus *Favia* were found as massive- and encrusting growth forms that experienced similar amounts of interaction with algal turfs. However, encrusting *Favia* colonies successfully outcompeted algal turfs more than twice as often as massive colonies (19.8% vs. 9.6% respectively) (Fig. 5B). Massive coral colonies within the genera *Favites* and *Montastrea* experienced 42.9% and 33.3% more interactions with turf algae respectively than encrusting colonies in the same genus (Figs. 5C and 5D). Within the genus *Montipora*, encrusting colonies had on average 2.45 times more interactions with turf algae along their edges than branching colonies.

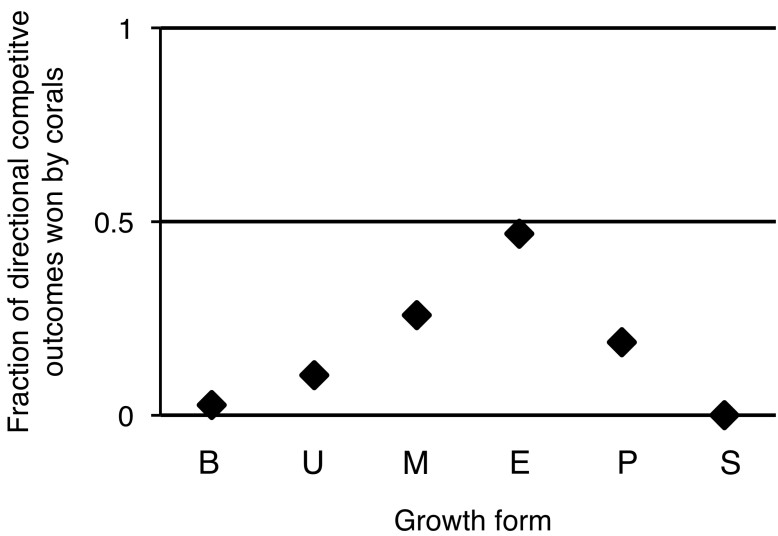

**Figure 4  Directional competitive outcomes per coral colony growth form.** Values are calculated by 'corals winning directional competitive interactions / total directional competitive interactions.' Abbreviations: B = Branching, U = Upright, M = Massive, E = Encrusting, P = Plating, S = Solitary.

However, the former outcompeted turf algae more often than the latter, winning 70.2% and losing 16.8% of the competitive interactions whereas the branching colonies were never observed winning an interaction and losing 86.8% of them (Fig. 5E). *Porites* was the most variable genus in terms of growth forms. Of the four growth forms we observed, encrusting colonies again experienced more competitive interaction along their borders (76.1%) than the other three growth forms and branching corals experienced the least (21.4%) relative to upright-(39.2%) and massive colonies (47.5%) (Fig. 5F). Encrusting corals again performed better in these interactions, winning 4.8 times more often than massive colonies and up to 55 times more often compared to branching colonies (Fig. 5F).

### Quantification and outcome of coral–algal interactions per coral colony size class

The relative percentage of a colony's border involved in competitive interactions with neighboring turf algae decreased as colonies increased in size (Table 3; Fig. 6A). Colonies larger than 80 cm experienced 7.5 times less competitive interactions with turf algae along their borders compared to the two smallest size classes (Fig. 6A). However, such differences could not be statistically supported, suggesting that growth form is foremost important in determining the outcome of competitive interactions with neighboring turf algae at our study site (Figs. 6B and 6C; Table 3).

### DISCUSSION

The amount and outcome of competitive interactions between corals and turf algae varied among coral growth forms. Encrusting corals experienced the highest amount of turf algal interaction along their perimeter compared to the other coral growth forms, but they were also more successful in competing against algal turfs. The opposite was observed for branching corals. While branching corals had a relatively low number branches covered

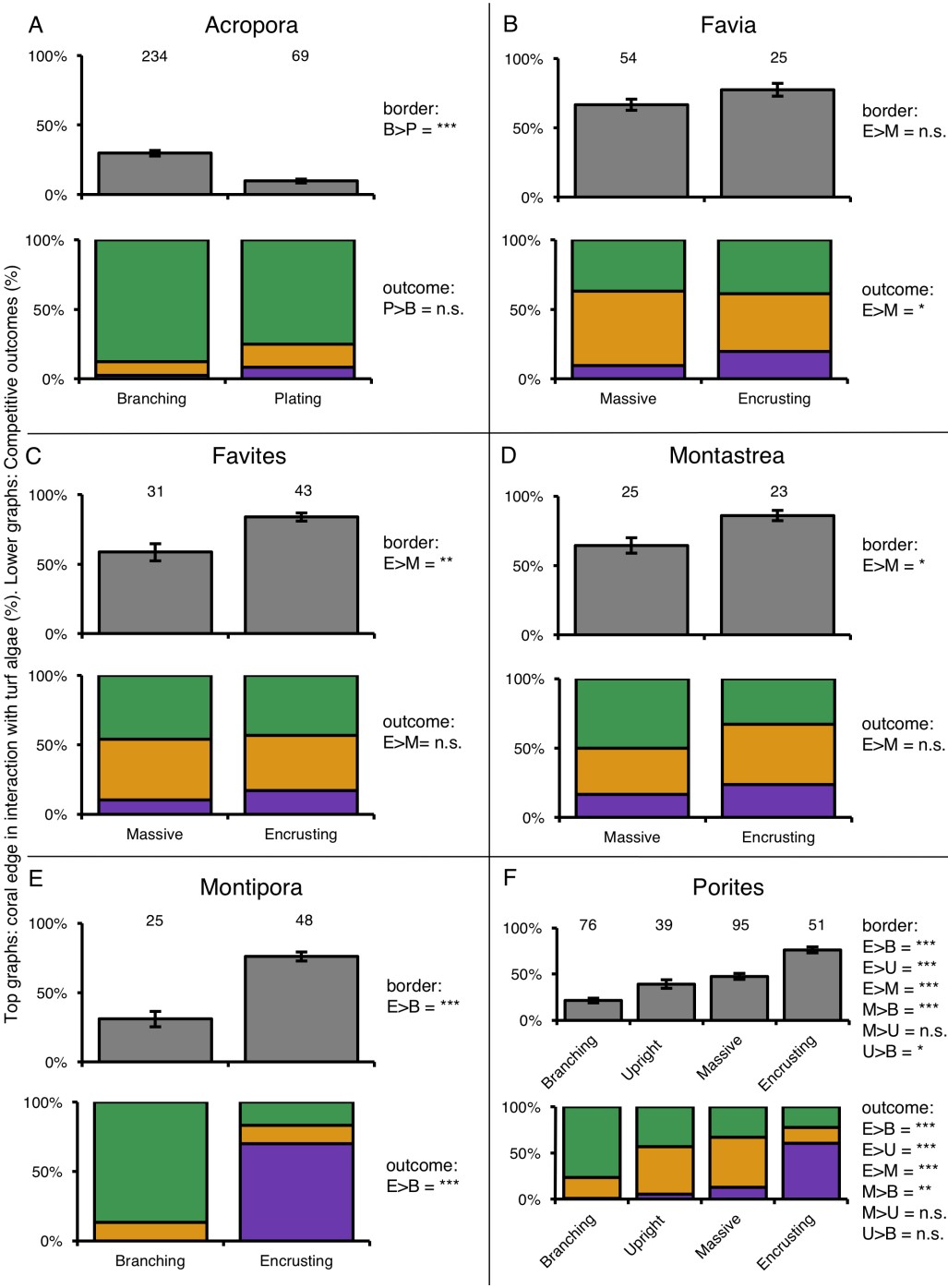

**Figure 5   Competitive interactions between different coral genera and turf algae per coral colony growth form.** Top graphs (grey color) indicate the amount of coral colony border interacting with turf algae. Lower graphs (colored) indicate the competitive outcomes of coral–turf algal interactions. Numbers above bars indicate the amount of samples. Significance levels of different quantities of coral–turf algal competition along coral borders (in row under 'Border') and of different competitive outcomes (in row under 'Competitive outcome') between the compered growth forms are stated on the right of the charts. The '>' means 'larger than'. *P*-values are indicated as $p < 0.05 = *$, $p < 0.01 = **$, $p < 0.001 = ***$, $p > 0.05 =$ n.s. (not significant). Abbreviations: B = Branching, E = Encrusting, M = Massive, P = Plating, U = Upright.

**Table 3  Pairwise comparisons between different coral colony size classes with the Kruskal–Wallis test.**
Levels of significance of comparisons between size classes for (I) different quantities of coral–turf algal
competition along coral borders (in column under 'Border') and (II) different competitive outcomes (in
column under 'Competitive outcome'). Size classes represent the maximum diameter of the coral colony
in centimeter, and the size classes are separated with a '/' (in column under 'Size classes').

| Size classes | Border | Competitive outcome |
|---|---|---|
| 0–5 cm / 5–10 cm | n.s. | n.s. |
| 0–5 cm / 10–20 cm | n.s. | n.s. |
| 0–5 cm / 20–40 cm | * | n.s. |
| 0–5 cm / 40–80 cm | *** | n.s. |
| 0–5 cm / 80+ cm | *** | n.s. |
| 5–10 cm / 10–20 cm | *** | n.s. |
| 5–10 cm / 20–40 cm | *** | n.s. |
| 5–10 cm / 40–80 cm | *** | n.s. |
| 5–10 cm / 80+ cm | *** | n.s. |
| 10–20 cm / 20–40 cm | n.s. | n.s. |
| 10–20 cm / 40–80 cm | *** | n.s. |
| 10–20 cm / 80+ cm | *** | n.s. |
| 20–40 cm / 40–80 cm | n.s. | n.s. |
| 20–40 cm / 80+ cm | n.s. | n.s. |
| 40–80 cm / 80+ cm | n.s. | n.s. |

Notes.
Asterisks indicate $p$-values after Bonferroni correction: $* \leq 0.05$; $** \leq 0.01$; $*** \leq 0.001$, n.s. = not significant.

with turf algae, they rarely won competitive interactions with turf algae. The other growth
forms ranked in between encrusting and branching corals, and growth forms experiencing
more competitive interaction from turf algae appeared better able to successfully compete
with these turf algae. The fact that similar results were found among and within coral
genera, shows that these differences should not only be attributed to genus-specific
responses to algal interactions but to growth form specific benefits involved in coral–turf
algal competitive interactions.

Corals can cope with turf algal competitive interactions in two different ways. First,
corals can use an 'escape in height' strategy (*Meesters, Wesseling & Bak, 1996*) establishing a
relatively small 'perimeter to surface area'-ratio which minimizes their exposure to nearby
benthic competitors. For example, plating corals have flat surfaces but grow slightly above
or over the bottom, thereby escaping interaction along the plate's growing edge. The base at
which the coral is attached to the benthos is heavily shaded by this plate, which drastically
decreases the abundance of light dependent turf algae. When algal turfs do manage to reach
the 'plate' of a colony, similar to the branches of branching corals, affected colonies are very
likely to lose this competitive interaction. Under the second strategy, corals do not invest
energy in vertical growth to avoid competitive interaction with turf algae altogether, but
actively fight off algae (*McCook, Jompa & Diaz-Pulido, 2001*) through abrasion, stinging,
allelopathy or mucus secretion (*Schoener, 1983*; *Lang & Chornesky, 1990*; *Karlson, 2002*).

Corals in this study appear capable of either quick growth to avoid benthic interaction
(e.g., plating and branching colonies) or slow growth in combination with defensive

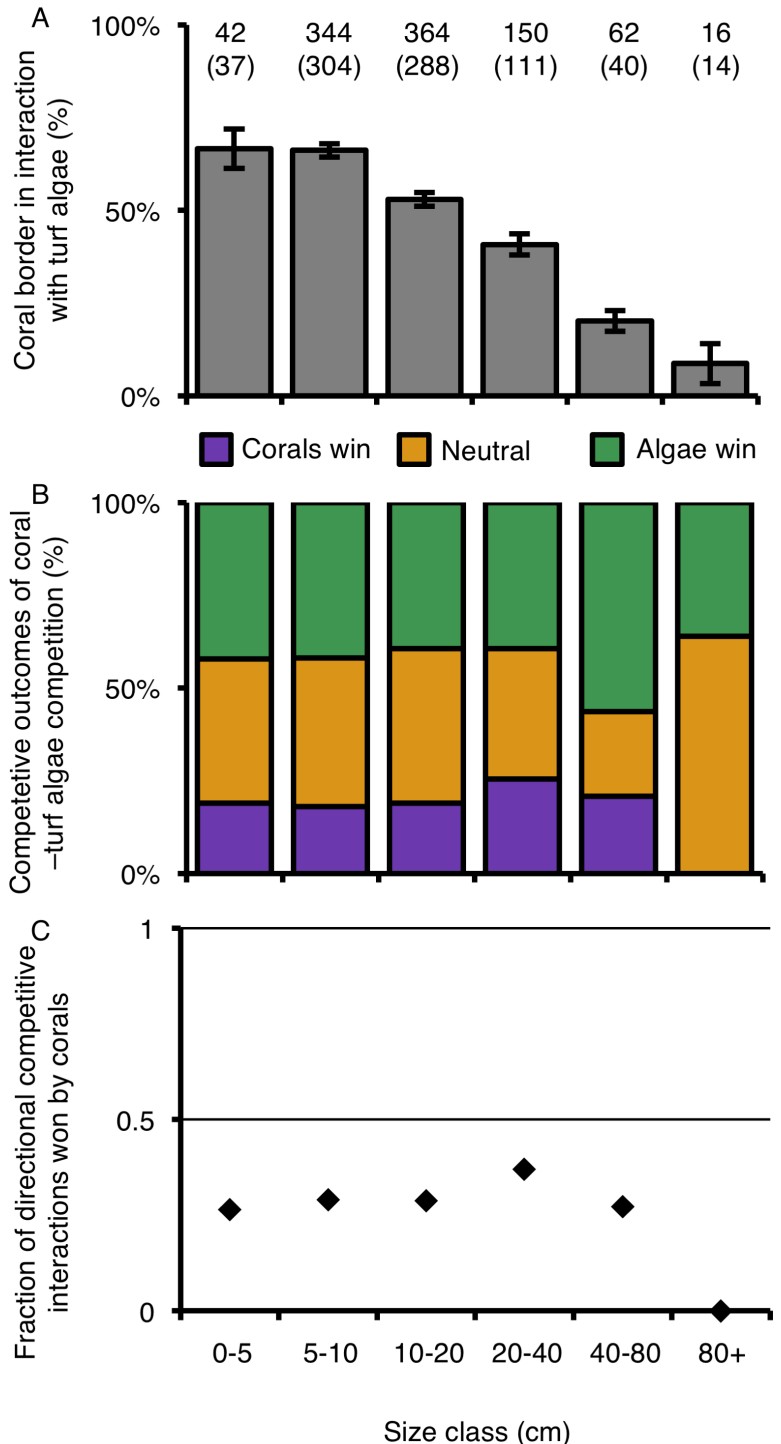

**Figure 6 Coral–turf algal interactions per size class.** (A) The amount of coral colony border interacting with turf algae per size class. (B) The competitive outcomes of interactions between corals and turf algae per size class. Purple indicates the proportion won by corals, green the proportion won by algae and orange the proportion of neutral interactions. (C) The directional competitive outcomes won by corals per size class. Numbers above graph A indicate the number of samples, numbers between brackets the number of corals involved in directional competitive interactions.

mechanisms (e.g., encrusting and massive species). This apparent trade-off between growth and investment in defenses has many similarities with the resource availability hypothesis (RAH), which states that fast growing, short-lived plant species invest less in defenses compared to slow growing long-lived species, since the relative costs of allocating resources away from growth to defensive mechanisms are higher for fast growing than for slow growing species (*Endara & Coley, 2011*). Our results strongly suggest that growth forms associated with fast growth (i.e., branching corals) are indeed less successful in competing with turf algae than growth forms associated with slow growth (i.e., encrusting-, massive corals). Ensuring that slow growing species indeed rely on active defense mechanisms to overtake or defend already occupied space within a reef community would be the next step to confirm the applicability of the RAH to corals (e.g., presence of sweeper tentacles, secondary metabolites). Furthermore, our findings were derived from Vietnamese reef communities and should be repeated for other regions before one can generalize our findings on the effects of coral growth forms on the outcome of competitive interactions with neighboring turf algae.

Shifts in community structure whereby certain growth forms survive stressful conditions better than others have been observed in Japan and resulted in a higher relative abundance of massive- and encrusting corals (*Loya et al., 2001*), confirming expectations following from our findings. The fact that a coral colony's performance in competitive interaction with algal turfs partially depends on its growth form implies that the composition and structural complexity of coral reefs is also likely to change in response to the observed increased presence of turf algae on reefs worldwide (*Gorgula & Connell, 2004*; *Hoegh-Guldberg et al., 2007*; *Hughes et al., 2007*). *Acropora* is a dominant Indo-Pacific coral genus (*Done, 1992*; *Connell et al., 2004*), and also the most abundant on our study site. *Acropora*'s have high susceptibility to bleaching (*Loya et al., 2001*; *Marshall & Baird, 2000*) and breakage during storms (*Muko et al., 2013*), which, in combination with their suboptimal performance while competing with turf algae (this study) makes branching *Acropora* corals especially vulnerable to this seemingly large variety of external stressors. A decrease in the abundance of branching corals would lead to a reduced three-dimensional structure of the reef, with far-reaching effects for other reef organisms, for example fishes, that depend on the shelter provided by the complex structures typical of branching corals (*Lirman, 1999*).

Our results only partially supported the conclusions from a similar study conducted by *Barott et al. (2012)* who found that small- and large-sized corals are better capable in fighting off algae than medium-sized corals. The authors suggested that small corals do not need to invest energy in reproduction, whereas medium sized coral do, and that large corals eventually benefit from the 'escape in height' strategy to avoid algal interaction. Our results indicated that with increasing coral colony size, the percentage of competitive interaction a colony experiences along its border tended to decrease, but no significant relationship could be found. Our findings therefore do not unequivocally support that large corals are better competitors against algal turfs, and suggest that they simply appear to be better in avoiding competitive interaction altogether.

This study contributes to our understanding of the relationships between coral growth forms and their competitive interactions with turf algae. General patterns were found that

transcend species identity which could help understand or quantify biological processes on highly biodiverse coral reef communities in the Indo-Pacific. Coral colony form, rather than size, proved a strong determinant to predict the outcome of competitive interaction between corals and neighboring turf algae.

## ACKNOWLEDGEMENTS

Many thanks to Six Senses resorts and the entire staff of Ninh Van Bay. Gary Hendon and Joana Sandkühler, thank you for providing all necessities to fulfill the field work and we admire your commitment to this research. Gratefulness is justified to Selma Ubels for helping with experimental design and sampling. The many discussions and support are well appreciated. Maarten van Gemert and Friso Dekker provided valuable advice during the analyses. Lastly, we thank Petra Visser for the feedback throughout all stages of the study.

### Funding

The authors received no funding for this work.

### Competing Interests

The authors declare there are no competing interests.

### Author Contributions

- Thomas Swierts conceived and designed the experiments, performed the experiments, analyzed the data, contributed reagents/materials/analysis tools, wrote the paper, prepared figures and/or tables, reviewed drafts of the paper.
- Mark JA Vermeij conceived and designed the experiments, contributed reagents/materials/analysis tools, wrote the paper, prepared figures and/or tables, reviewed drafts of the paper.

### Data Availability

All data is provided as Supplemental Information.

### Supplemental Information

Supplemental information for this article can be found online at http://dx.doi.org/10.7717/peerj.1984#supplemental-information.

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
