# Peer review of "Competitive interactions between corals and turf algae depend on coral colony form"

_PeerJ, doi:10.7717/peerj.1984_

## Round 0.1 · original submission · Major Revisions

In general the reviewers found the study of interest.The rReviewers raised the same concern, in different degrees, about the temporal approach (snapshot) used to assess the physical interaction between coral and algae, and have provided some recommendations to re-approach interpretations and re-phrase the title.

The authors need to review some of the terminology used throughout the paper such as “competition” versus “interaction”. Clarification is also needed on several methodological aspects raised (in detail) by all reviewers. There is an agreement that more clarification is needed on how the interaction outcomes were assessed/gauged (coral damaging algae, algae damaging coral and neutral).

Two of the reviewers (#1 and #3) raised strong concerns of the approach of how some branching corals, such as Acropora, were assigned to some atypical morphological categories. Therefore it requires revision.

All the raised points need to be addressed before we can make a decision on publication.

Reviewer 1 ·

Basic reporting

The study depicted the percentage area of coral interacting with turf algae in a Vietnamese reef site. The study also showed an effect of coral morphology on the amount of coral area interacting with turf algae. Coral morphology may affect the outcome of these interactions, as snapshots in time resulted in encrusting forms winning encounters with turf algae more often than branching corals did (43 vs 3% of encounters). As morphology types were examined regardless of coral species, the results highlighted the importance of growth forms in the coral – turf algae interactions.
The ms would benefit from clarification of terms such as "outcome", “size and the amount of turf algal competition”, “degree of turf algal competition”, “highest degree of algal competition”. Also, the ms length can be shortened by eliminating Figures and/or Tables and it can be streamed by avoiding unnecessary text and/or analyses. For example, I consider that the analysis by genus is not well justified (Fig 5) and I would suggest to eliminate it. Also Figure 2 is nice and can be helpful but is not essential.
The authors stressed difficulties in assessing competition outcome from single time observations (L157). However, the way in which some results are presented and the title of the ms are misleading. I would suggest to make the necessary changes along the ms, and modify the title, such as: “Number and type of competitive interactions between corals and turf algae depend on coral colony form” or simply “competitive interactions between corals and turf algae depend on coral colony form”.

Please find detailed comments below:
Abstract
L34-38: “In total, the size and the amount of turf algal competition was quantified…”; it is confusing. It is unclear what was measured. Also, “the degree of turf algal competition” and “the highest degree of algal competition” are unclear. Please define size, amount and degree of competition.
L41-45: Apparently, the message of the last sentence is that coral ability to outcompete turf algae (TA) varied within genus among 6 genera examined. Each genus, includes a variety of growth forms and many species, thus the significance of this result or the message intended is unclear to me. Please re-write.

Introduction
L88-92: this last sentence is too long to follow. Please split in two.
L95: double parentheses
L115: experience “less competition”; phrasing seems incorrect; probably more appropriate to use experience less antagonistic behavior; less interacting surface (or perimeter) compared to the total surface (or perimeter) of the organism or something alike.

Methods
L135: please include a depth range for transect deployment in the reef
L143: I suggest to modify this sentence as: The only exception to this method was for branching Acropora corals since colony basis were hidden under an entanglement of branches.
L145: These tangled bunches of Acropora could have more than 100 branches?
L148-149: in such entanglements of branches it is very difficult to distinguish the primary branch; how do you solve this?
L155: How do you account for multiple interactions in a coral within the analysis?
L158:…and that the actual trend of….
L169: the solitary corals may include attached and non-attached colonies; it can be useful to mention why you limit your survey to non-attached species (eg because only Fungiidae is present at your site). Also, “solitary” does not seem to fall within a specific coral morphology but it may be argued the contrary. Some Fungia and Ctenactis species can be considered as “massive” in their growth form.

Results
L199; this reef only have stony corals, macroalgae and TA? Please explain “the remaining 15.7% consisted of non-biological substrates”; no sponges, bryozoans, octocorals?
L206: 1096 colonies + 175 Acropora colonies = 1271 vs 1276 in the Abstract; please check
L208: “ most” not necessary before dominant
L215: pls consider a substitute for “suffered” (again in L221)

Experimental design

The methods used to accommodate big colonies of branching Acropora seem drastically different to those used for the rest of the colonies. This difference in method requires a better justification to include the branching Acropora colonies in the analysis. Difficulties in determining the boundaries of a colony for big branching Acropora, prompted the authors to exclude them from the size class comparisons (L173-175). For this same reason, I would prefer to see big Acropora colonies excluded and to see competition results among colony morphologies that were assessed with a similar method. Apparently, these big branching Acropora only represent a relatively small fraction of the total number of colonies (175 out of 1271).
The percentage of coral area that is interacting with turf algae (TA) varied in relation to coral morphology. Then, it is unclear why the potential effect of coral size in coral-TA interactions was analyzed for all growth forms grouped (Fig 6 & Table 3). I think that this effect would be better assessed by analyzing different size classes within each morphology.

Table 1: p-values indicate sig levels of the disproportionality between the quantities of the TA along corals compared to its cover on the reef. Please explain which test was used and what was considered as a replicate.
Table 3: What is the first column? I did not understand “significant levels of different quantities of coral-algal competition along coral borders”; Is it coral vs algal size classes? Please explain

Validity of the findings

Please see previous general comments.

Additional comments

I consider that the study makes a contribution to the understanding of competition between corals and turf algae by supporting the results as expressed in the first paragraph under "Basic Reporting". Therefore, I encourage you to address these comments, made to help improving the ms.

Reviewer 2 ·

Basic reporting

No comments

Experimental design

* The Resource Availability Hypothesis for different growth forms of coral in competition with turf algae could not be tested because as the authors stated they took a snapshot of the interactions. A snapshot is not able to tell us if the outcome was caused by algae actively overgrowing live corals or passively taking over space after corals have died.
* The different methods used to measure interaction with turf algae probably played a big role on the results obtained. When measuring interaction not on the border of branching corals but on the branches, authors assumed that turf algae get to the branch as easy as to the borders of the other growth forms. The number are not measuring the same thing and cannot be compared.
* Methods to determine the putative outcome were not enough explained and cannot be reproducible. For example, what would be considered the final competitive outcome if in one end of a colony the coral was overgrown by algae but at the other end the coral made a huge advance towards the algae? Methods from Barott et al. (2012b) are also incomplete.
* Authors said: To compare growth forms within a single genus, growth forms had to be represented by at least 20 individuals.
Why 20? What if were there only very few colonies out of the 20 colonies in one growth form category? Was the analysis still performed?

Validity of the findings

The conclusion is not supported by the results. The results say more about mortality than competition. Authors are assuming mortality was caused by competition.

Additional comments

In case methods can be better explained, I encourage authors to rewrite their manuscript focusing on differences in coral mortality of different coral growth forms. Competition and RAH Hypothesis should be discussed as speculations only.
I have attached a PDF file with some comments and suggestions for improvement.

Annotated reviews are not available for download in order to protect the identity of reviewers who chose to remain anonymous.

·

Basic reporting

I am generally impressed with this work and how it is presented.

I have minor reservations regarding:

The use of the word competition where in some instances the word interaction or contact may be more appropriate. This is because I interpret the use of competition in this type of study to imply an interaction that has been observed over time to demonstrate a competitive effect of one organism or assemblage on another. The authors appear to acknowledge this on lines 157-159.

I am concerned that lumping coral morphologies into the category of ‘upright’ may create some confusion relative to previous studies and believe use of standard groupings defined by for example Veron (2000) may be more appropriate.

On lines 68-9 the opening sentence suggests that turf algae are not macroalgae. I believe this is confusing as turf algae are often a mix of diminutive macroalgae together with filamentous macroalgae. Nonetheless, I believe filamentous algae are also macroalgae as opposed to microalgae as defined for example by Steneck & D’ethier (1994).

On lines 75-78 I believe it is also appropriate to mention that the physical features of a turf assemblage, such as height and density are also relevant to competitive interactions.

On line 183 there appears to be a typo where “lasts” should be “last”

There appears to be some inconsistent use of present and past tense in the manuscript. Contrast for example lines 206-207 with lines 212-214.

The heading at line 238 refers to genera and therefore should be changed to “taxa specific” instead of “species specific”.

Experimental design

Regarding the approach for branching coral described from line 145-149. To convince the reader of this approach it would benefit from some basic calibration, where actual field observations are compared to the interpretation of analysed photographs.

In the definition of the categories used for directional outcomes of competition, won, lost and neutral from line 151 it is not clear exactly what observations were required by the authors to define a win, lose or neutral outcome. Given that the interactions are not followed over time it cannot be ascertained from a change in surface area of coral versus turf algae. Therefore as an example, I assume the authors have used features such as coral tissue necrosis or a bleached aspect of the turf algae as an indication. I think it would be good to define this approach more specifically.

In the definition of coral morphologies starting at line 162, I believe that it is somewhat confusing to the literature to rename colony growth form categories (e.g. relative to Veron 2000), although I am not adverse to the lumping morphologies for analysis.

Validity of the findings

I agree with the interpretation of the results and believe this work makes progress in the field, whilst also draws attention to the need for future work adressing specifics of coral-algal interactions.

Additional comments

I believe this is an interesting piece of work that makes a good contribution to the current literature.

---

## Round 0.2 · Minor Revisions

Your resubmission has been reviewed. Some minor rewording and editorial changes have been suggested and I would encourage you to address them. Regarding experimental design, the reviewer has brought up two points that need your consideration, which I think might impact/help in the power of your analyses.

Reviewer 1 ·

Basic reporting

A sense of time (short-term) and possibly of space (a reef in the South China Sea) should be included in the Title of the ms. This suggestion arises as this is a study focused on short-term interactions, where reversibility of competing outcomes are likely (eg Chornesky, E. A. (1989). Repeated reversals during spatial competition between corals. Ecology, 70(4), 843–855; Aliño, P. M., Sammarco, P. W., & Coll, J. C. (1992). Competitive strategies in soft corals (Coelenterata, Octocorallia). IV. Environmentally induced reversals in competitive superiority. Marine Ecology Progress Series, 81, 129–145.)

Editorial suggestions:
Abstract: define “most successful competitor”; is it predictably or consistently?
Introduction: L58, Delete “in short” on second sentence. L89, delete “are” before “therefore”
Table 2 caption: correct “compared”; in my view, wording in Table 2 and 3 captions is still difficult to follow.

Experimental design

M&M: L160-162, Please explain in more detail how multiple interactions in a single coral colony were assessed, converted proportionally to different outcomes and included in the analyses.

Comments to concern 1.24 in rebuttal letter: In addition to increase the number of replicates, one way around this lack of power might be to reduce the number of size categories or morphologies or both. Did you try this?

Validity of the findings

L326-327: “Coral colony form, rather than size, proved a strong determinant to predict the outcome of competitive interaction…”; Again, due to the lack of integrated comparison of these two factors (form and size) in a single analysis, I would suggest to reword this sentence of last paragraph in Discussion.

Additional comments

Thanks for addressing most of my comments to the first revision.

---

## Round 0.3 · accepted · Accept

All reviewer's comments and suggestions have been addressed satisfactorily by the authors.